

# Mortality in HIV-infected patients with tuberculosis treated with streptomycin and a two-week intensified regimen: data from an HIV cohort study using inverse probability of treatment weighting

Gerardo Alvarez-Uria, Manoranjan Midde and Praveen K. Naik

Department of Infectious Diseases, Rural Development Trust, Bathalapalli, AP, India

## ABSTRACT

**Background.** Despite the dramatic scale-up of antiretroviral therapy in low- and middle-income countries, tuberculosis (TB) is still the main cause of death among HIV-infected patients in resource-limited settings. Previous studies in patients with TB meningitis suggest that the use of higher doses of common anti-TB drugs could reduce mortality.

**Methods.** Using clinical data from an HIV cohort study in India, we compared the mortality among HIV-infected patients diagnosed with TB according to the regimen received during the first two weeks of treatment: standard anti-tuberculosis therapy (ATT) ($N = 847$), intensified ATT ($N = 322$), and intensified ATT with streptomycin ($N = 446$). The intensified ATT comprised double dose of rifampicin and substitution of ethambutol with levofloxacin. Multivariate analysis was performed using Cox proportional hazard models and inverse probability of treatment weighting (IPTW) based on propensity scores. Patients with TB meningitis were excluded.

**Results.** The use of intensified ATT alone did not improve survival. However, when streptomycin was added, the use intensified ATT was associated with reduced mortality in Cox models (adjusted hazard ratio 0.72, 95% CI [0.57–0.91]) and after IPTW (hazard ratio 0.77, 95% CI [0.67–0.96]). Other factors associated with improved survival were high serum albumin concentration, high CD4 lymphocyte cell-counts, and high glomerular filtration rates. Factors associated with increased mortality were high urea concentrations, being on antiretroviral therapy at the time of ATT initiation and high BUN/creatinine ratio. In an effect modification analysis, the survival benefits of the intensified ATT with streptomycin disappeared in patients with severe hypoalbuminemia.

**Conclusion.** The results of this study are in accordance with a previous study from our cohort involving patients with TB meningitis, and suggest that an intensified 2-week ATT with streptomycin could reduce mortality in HIV infected patients with TB. As this is an observational study, we should be cautious about our conclusions, but given the high mortality of HIV-related TB, our findings deserve further research.

Corresponding author
Gerardo Alvarez-Uria,
gerardouria@gmail.com

## INTRODUCTION

Tuberculosis (TB) accounts for one third of deaths among HIV-infected people, and HIV infection is present in 25% of TB deaths (*World Health Organization, 2015*). Globally 1.2 million people living with HIV had TB in 2014, and 390,000 people died (*World Health Organization, 2015*). Despite this tremendous burden of preventable deaths, there has not been any major breakthrough in the chemotherapy of HIV-related TB in the last decades.

Recent data from a clinical trial and an observational cohort study suggest that increasing the doses of common anti-TB drugs during the initial phase of anti-TB therapy (ATT) could have survival benefits in patients with TB meningitis (*Alvarez-Uria et al., 2013a*; *Ruslami et al., 2013*). In a phase two randomized trial investigating the effect of an intensified ATT (iATT) during the first two weeks of treatment, higher exposure to rifampicin was associated with improved survival (*Ruslami et al., 2013*; *Te Brake et al., 2015*). In previous studies from our HIV cohort, an iATT during the first days of TB meningitis treatment also achieved improved survival, but only if streptomycin (STM) was added to the regimen (*Alvarez-Uria et al., 2015*). This finding was intriguing because STM has poor penetration into cerebrospinal fluid (*Donald, 2010*), suggesting that the beneficial result of the regimen was not related to the local effect in CSF.

Data of intensified regimens in HIV-related TB other than TB meningitis are lacking. In the present study, we hypothesized that an iATT could also reduce mortality in other forms of TB. We aimed to compare the survival of HIV-infected patients with TB treated with a standard ATT (sATT) and an iATT (with and without STM) during the first two weeks of chemotherapy.

## METHODS

### Setting and design

The Vicente Ferrer HIV Cohort Study (VFHCS) is an open cohort study of HIV-infected patients who have attended the Rural Development Trust Hospital in Bathalapalli, Anantapur District, AP, India. The hospital provides medical care free of charge to people living with HIV. In our setting, 72% of the population live in rural areas (*Office of The Registrar General & Census Commissioner, India, 2011*). The HIV epidemic is largely driven by heterosexual transmission and it is characterized by low CD4 cell counts at presentation, poor socioeconomic conditions and high levels of illiteracy (*Alvarez-Uria et al., 2012a*; *Alvarez-Uria et al., 2012b*; *Alvarez-Uria et al., 2013c*). The VFHCS is registered at ClinicalTrials.gov (number NCT02454569).

For this study, we included all HIV infected patients diagnosed with tuberculosis from 1 June 2012 to 12 May 2015 from the VFHCS database. The selection of patients from the database was executed on 16 November 2015. Patients with tuberculous meningitis were excluded. There were no other exclusion criteria.

The selection of variables included in the multivariate analysis was performed based on previous studies investigating risk factors for mortality (*Alvarez-Uria et al., 2012a*; *Alvarez-Uria et al., 2013b*; *Alvarez-Uria et al., 2014a*; *Alvarez-Uria et al., 2014b*; *Kim et al., 2012*; *Baghaei et al., 2014*). Designation of the community of patients was performed by

self-identification. Scheduled caste community is marginalised in the traditional Hindu caste hierarchy and, therefore, suffers social and economical exclusion and disadvantage (*Gang, Sen & Yun, 2002*). Scheduled tribe community is generally geographically isolated with limited economical and social contact with the rest of the population (*Gang, Sen & Yun, 2002*). Scheduled castes and scheduled tribes are considered socially disadvantaged communities and are supported by positive discrimination schemes operated by the Government of India (*Alvarez-Uria, Midde & Naik, 2012*). The estimated glomerular filtration rate (EGFR) was calculated using the Chronic Kidney Disease Epidemiology Collaboration (CKD-EPI) equation (*Teo et al., 2011*; *Lucas et al., 2014*). Mild extrapulmonary TB was defined in patients diagnosed with TB pleuritis or peripheral lymphadenitis (but not other forms of extrapulmonary TB) and no acid-fast bacilli in sputum (*Alvarez-Uria et al., 2012a*).

## Diagnosis of TB

Acid fast bacilli staining of sputum and chest radiograph were performed for all patients with a clinical suspicion of TB. Analysis of cerebrospinal fluid, pleural fluid or ascitic fluid was performed if there were signs of neurological involvement, pleural fluid in the chest radiograph or ascites, respectively. In smear-negative patients complaining of important weight loss, an abdominal ultrasound was performed to search for signs of abdominal TB (*Tarantino et al., 2003*; *Sharma et al., 2007*).

The diagnosis of TB was made according to World Health Organization recommendations for the definition of TB case and the locally available standard of care (*World Health Organization, 2010*). Thus, TB diagnosis was based on the presence of acid fast bacilli on sputum smear, caseating or necrotizing granuloma in clinical specimens, or clinical presentation suggestive of TB along with supportive findings in the chest radiograph, abdominal ultrasound or laboratory results from biological fluids.

## Treatment during the first two weeks of ATT

Patients were divided into three treatment groups following an intention to treat analysis. Patients in the sATT group received isoniazid 300 mg, rifampicin 450 mg, ethambutol 800 mg, and pyrazinamide 1,500 mg. Patients in the iATT group received isoniazid 300 mg, rifampicin 900 mg, pyrazinamide 1,500 mg and levofloxacin 1,000 mg. In patients from the iATT + STM group, STM 750 mg (intramuscular) was added to the iATT regimen. Co-trimoxazole prophylaxis was given to all patients. All patients were admitted to the hospital and those not on ART at the time of ATT initiation were counselled to start ART between 2–8 weeks after hospital discharge.

## Statistical analysis

We used time-to-event methods to study the mortality during the first twelve months after starting ATT. Time was measured from ATT initiation to death. Patients who did not die during the study period were censored at twelve months or at their latest visit date, whichever occurred first. Univariate and multivariate analyses were performed using Cox proportional hazard models. The proportional hazard assumption was assessed performing log–log survival curves based on Schoenfeld residuals (*Kleinbaum & Klein, 2005*). The log-linearity assumption was checked for all continuous variables, and CD4 + lymphocyte concentration in blood had to be log transformed to achieve log-linearity.
Furthermore, we performed an additional analysis using inverse probability of treatment weighting (IPTW) (*McCaffrey et al., 2013*). To minimize the effect of confounding and obtain an unbiased estimate of the treatment effects, the strongest predictors of mortality were balanced using propensity score methods. The strongest predictors of mortality were selected parsimoniously using a backward elimination method that roughly imitates selection according to a minimal Akaike's information criterion (*Royston & Sauerbrei, 2008*). Propensity scores were estimated via boosted models using the "twang" package in the R statistical computing environment (R Foundation for Statistical Computing, Vienna, Austria), which takes into account non-linear effects and interactions (*Ridgeway et al., 2013*). To select the optimal interation of generalized boosted models, we set to minimize the means of absolute standardized differences (*McCaffrey et al., 2013*). The propensity scores were used to estimate stabilized weights (*McCaffrey et al., 2013*). These sampling weights were used to compare the mortality of the sATT group and the iATT + STM group using robust variance to account for the weighted nature of the sample (*Austin, 2013*). To investigate treatment effect modification, all two-way interactions were assessed using multivariate fractional polynomials (*Royston & Sauerbrei, 2009*; *Royston & Sauerbrei, 2014*).

Except for the estimation of propensity scores, statistical analysis was performed using Stata Statistical Software (Stata Corporation. Release 13.1. College Station, Texas, USA).

The VFHCS was performed according to the principles of the Declaration of Helsinki, and was approved by the Ethics Committee of the Rural Development Trust Hospital (Reference number OS/003). Written informed consent was given by patients or caretakers for their information to be stored in the study database and used for research.

## RESULTS

During the study period, 1,615 patients started ATT; 847 were included in the sATT group, 322 in the iATT group, and 446 in the iATT + STM group. Baseline characteristics of patients by treatment group are presented in  Table 1. The proportion of patients previously treated of tuberculosis was higher in the iATT + STM group (43%) than the one in the sATT group (19.2%) and in the iATT group (none). This is because the Indian Guidelines for tuberculosis recommend adding STM during the first two months of ATT in patients who had received ATT for at least one month in the past (*Ministry of Health and Family Welfare India, 2005*).

Patients in the sATT group had higher concentrations of serum albumin, higher EGFRs and lower blood urea nitrogen (BUN)/creatinine ratios. Other differences were not statistically significant. Weight was measured in 780 patients in the sATT group, 289 patients in the iATT group and 401 patients in the iATT + STM group, and the median weight was 47 kg (interquartile range (IQR), 40–55), 47 kg (IQR, 40–55), and 45 kg (IQR, 40–52), respectively. Thus, the median dose of rifampicin was 9.6 mg/kg (IQR, 8.2–11.3) in the sATT group, 19.1 mg/kg (IQR, 16.4–22.5) in the iATT group and 20 mg/kg (IQR, 17.3–22.5) in the iATT + STM group.

Univariate and multivariate analysis of factors associated with mortality are presented in Table 2. After adjusting for other covariates, there were no mortality differences

**Table 1  Baseline characteristics by treatment group.**

| | Standard ATT (n = 847) | Intensified ATT (n = 322) | Intensified ATT + STM (n = 446) | p-value |
|---|---|---|---|---|
| Gender | | | | 0.183 |
| Male | 557 (65.76) | 203 (63.04) | 309 (69.28) | |
| Female | 290 (34.24) | 119 (36.96) | 137 (30.72) | |
| On ART | | | | 0.429 |
| No | 576 (68) | 231 (71.74) | 303 (67.94) | |
| Yes | 271 (32) | 91 (28.26) | 143 (32.06) | |
| Disadvantaged community | | | | 0.796 |
| No | 579 (68.36) | 220 (68.32) | 297 (66.59) | |
| Yes | 268 (31.64) | 102 (31.68) | 149 (33.41) | |
| Homeless | | | | 0.786 |
| No | 831 (98.11) | 317 (98.45) | 436 (97.76) | |
| Yes | 16 (1.89) | 5 (1.55) | 10 (2.24) | |
| Previous ATT | | | | < 0.001 |
| No | 684 (80.76) | 322 (100) | 254 (56.95) | |
| Yes | 163 (19.24) | 0 (0) | 192 (43.05) | |
| Mild extrapulmonary TB | | | | 0.29 |
| No | 688 (81.23) | 252 (78.26) | 369 (82.74) | |
| Yes | 159 (18.77) | 70 (21.74) | 77 (17.26) | |
| Age (years), median (IQR) | 36.07 (30.02–44.42) | 34.995 (29.98–44.97) | 36.28 (31.19–43.97) | 0.43 |
| Albumin (g/dl), median (IQR) | 3.2 (2.7–3.7) | 3.15 (2.7–3.6) | 3.1 (2.6–3.5) | 0.0013 |
| CD4 count (cells/μl), median (IQR) | 116 (59–222) | 102 (49–210) | 118 (56–242) | 0.16 |
| Urea (mg/dl), median (IQR) | 23.5 (17.3–34.9) | 23.6 (16.5–35.1) | 23.3 (17–35.7) | 0.9 |
| BUN/Creatinine ratio, median (IQR) | 14.23 (10.67–19.19) | 16.71 (12.76–22.4) | 16.92 (12.4–22.59) | < 0.001 |
| EGFR (mL/min/1.73 m²), median (IQR) | 109 (86–122) | 118 (98–133) | 118.5 (100–134) | < 0.001 |

**Notes.**

ART, Antiretroviral therapy; ATT, Anti-tuberculosis therapy; BUN, Blood urea nitrogen; EGFR, Estimated filtration rate; IQR, Interquartile range; STM, Streptomycin.

Data are presented as No. (%) unless otherwise indicated. P-values were calculated using Chi² test for categorical variables and Kruskal-Wallis rank test for continuous variables.

between the sATT and the iATT groups, but the use of iATT + STM versus sATT was associated with a 28% mortality risk reduction (adjusted hazard ratio [aHR] 0.72, 95% CI [0.57–0.91]) compared with sATT. Other factors associated with mortality were low albumin concentrations (aHR 0.47 per increase of 1 g/dl, 95% CI [0.41–0.54]), low CD4 + lymphocyte concentrations (aHR 0.76 per increase of 1 log-CD4 cells, 95% CI [0.69–0.83]), high urea concentrations (aHR 1.008 per increase of 1 mg/dl, 95% CI 1[.004–1.013]), high BUN/creatinine ratios (aHR 1.02, 95% CI [1.01–1.031]) and lower EGFR (aHR 0.994 per increase of 1 ml/min/1.73 m², 95% CI [0.99–0.999]). Having mild extrapulmonary TB was associated with reduced risk of death (aHR 0.58, 95% CI [0.43–0.78]).

In a sensitivity analysis, we compared only the sATT group and iATT + STM group using IPTW. Variables statistically associated with mortality in the multivariate analysis were balanced using propensity score methods. At baseline, the sATT group had higher mean of serum albumin concentration, lower mean of EGFR, and lower mean of BUN/Creatinine ratio than the iATT + STM group (Table 3). These differences were considerably reduced

**Table 2** Univariate and multivariate analyses of risk factors for mortality using Cox proportional hazard models.

| | HR (95% CI) | Adjusted HR (95% CI) |
|---|---|---|
| Female | 1.030 (0.857–1.238) | 1.046 (0.856–1.277) |
| On ART | 1.038 (0.860–1.252) | 1.333* (1.088–1.633) |
| Disadvantaged community | 0.917 (0.759–1.108) | 0.871 (0.718–1.057) |
| Homeless | 1.750* (1.064–2.880) | 1.480 (0.895–2.449) |
| Previous ATT | 1.120 (0.912–1.376) | 1.163 (0.917–1.474) |
| Mild extrapulmonary TB | 0.418* (0.312–0.560) | 0.581* (0.432–0.781) |
| Age (years) | 1.010* (1.002–1.019) | 0.998 (0.988–1.009) |
| Albumin (g/dl) | 0.406* (0.356–0.461) | 0.471* (0.411–0.540) |
| Log-CD4 count (cells/$\mu$l) | 0.652* (0.602–0.705) | 0.759* (0.694–0.830) |
| Urea (mg/dl) | 1.017* (1.015–1.019) | 1.008* (1.004–1.013) |
| BUN/Creatinine ratio | 1.043* (1.035–1.050) | 1.020* (1.010–1.031) |
| EGFR (mL/min/1.73 m$^2$) | 0.990* (0.987–0.993) | 0.994* (0.990–0.999) |
| ATT | | |
| Standard | 1 (Reference) | 1 (Reference) |
| Intensified | 0.952 (0.758–1.195) | 0.868 (0.682–1.103) |
| Intensified + STM | 0.837 (0.674–1.040) | 0.718* (0.569–0.907) |

**Notes.**

*$P$-value $< 0.05$.

ART, antiretroviral therapy; ATT, anti-tuberculosis therapy; BUN, blood urea nitrogen; EGFR, estimated filtration rate; HR, hazard ratio; STM, streptomycin.

after IPTW. The median of propensity scores was 0.3 (IQR 0.25–0.37) in the sATT group and 0.39 (IQR, 0.32–0.51) in the iATT + STM group. The mean standardized difference and the mean Kolmogorov–Smirnov statistic was 0.111 and 0.064, respectively before IPTW, and 0.040 and 0.029, respectively after IPTW. The maximum standardized difference and the maximum Kolmogorov–Smirnov statistic was 0.349 and 0.192, respectively before IPTW, and 0.0858 and 0.058, respectively after IPTW.

Stabilized sampling weights were used to estimate Kaplan–Meier survival curves and Cox proportional hazards. In Fig. 1, we present the Kaplan–Meier survival estimates. The use of iATT + STM was associated with a significant reduction in mortality (HR 0.766, 95% CI [0.67–0.96]). While analysing interactions between treatment and other predictors of mortality, we found that baseline serum albumin concentrations modified significantly the survival effects seen with the iATT + STM regimen ($P$-value for interactions = 0.0003). Compared with sATT, the iATT + STM regimen was more beneficial for patients with higher serum albumin concentrations than for those with more severe hypoalbuminemia (Fig. 2).

## DISCUSSION

HIV-associated TB is difficult to treat. According to the latest Global TB Report (*World Health Organization, 2015*), nearly one third of HIV-infected patients with TB die with the current standard of care.

**Table 3   Balance before and after inverse probability of treatment weighting.**

| | Mean (sATT) | Standard deviation (sATT) | Mean (iATT + STM) | Standard deviation (iATT + STM) | Standardized difference | *P*-value | KS statistic | KS *P*-value |
|---|---|---|---|---|---|---|---|---|
| BEFORE INVERSE PROBABILITY OF TREATMENT WEIGHTING | | | | | | | | |
| Log-CD4 count (cells/μl) | 4.693 | 1.001 | 4.691 | 1.063 | − 0.002 | 0.969 | 0.042 | 0.653 |
| Albumin (g/dl) | 3.214 | 0.743 | 3.051 | 0.652 | − 0.227 | 0 | 0.106 | 0.002 |
| On ART | 0.32 | 0.466 | 0.321 | 0.467 | 0.001 | 0.98 | 0.001 | 0.98 |
| Mild extrapulmonary TB | 0.188 | 0.39 | 0.173 | 0.378 | − 0.04 | 0.505 | 0.015 | 0.505 |
| EGFR (mL/min/1.73 m$^2$) | 103.609 | 29.709 | 114.334 | 31.354 | 0.349 | 0 | 0.192 | 0 |
| BUN/Creatinine ratio | 15.962 | 8.015 | 18.873 | 9.521 | 0.336 | 0 | 0.176 | 0 |
| Urea (mg/dl) | 29.958 | 22.03 | 30.034 | 22.924 | 0.003 | 0.954 | 0.03 | 0.945 |
| AFTER INVERSE PROBABILITY OF TREATMENT WEIGHTING | | | | | | | | |
| Log-CD4 count (cells/μl) | 4.693 | 1.009 | 4.66 | 1.06 | − 0.032 | 0.608 | 0.03 | 0.955 |
| Albumin (g/dl) | 3.158 | 0.724 | 3.099 | 0.657 | −0.086 | 0.149 | 0.058 | 0.298 |
| On ART | 0.32 | 0.466 | 0.315 | 0.465 | − 0.01 | 0.871 | 0.005 | 0.871 |
| Mild extrapulmonary TB | 0.185 | 0.388 | 0.173 | 0.378 | − 0.032 | 0.597 | 0.012 | 0.597 |
| EGFR (mL/min/1.73 m$^2$) | 106.719 | 30.375 | 108.737 | 30.808 | 0.066 | 0.275 | 0.056 | 0.345 |
| BUN/Creatinine ratio | 16.658 | 8.394 | 17.34 | 8.429 | 0.081 | 0.161 | 0.054 | 0.39 |
| Urea (mg/dl) | 29.89 | 21.778 | 30.228 | 23.54 | 0.015 | 0.809 | 0.028 | 0.978 |

**Notes.**
ART, Antiretroviral therapy; iATT, Intensified anti-tuberculosis therapy; sATT, Standard anti-tuberculosis therapy; BUN, Blood urea nitrogen; EGFR, Estimated filtration rate; KS, Kolmogorov–Smirnov; STM, Streptomycin.

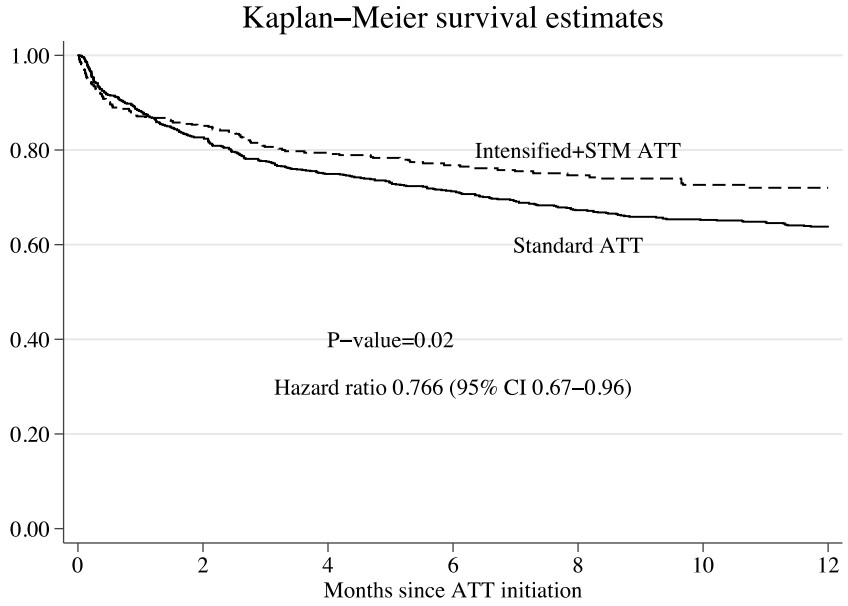

**Figure 1   Survival curves after inverse probability of treatment weighting.** ATT, anti-tuberculosis therapy; STM, streptomycin.

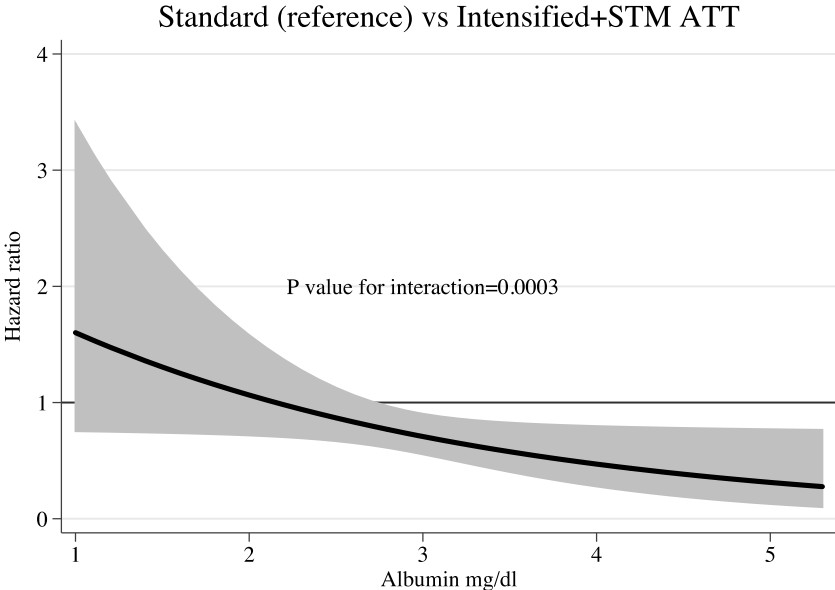

**Figure 2** **Hazard ratio and 95% confidence interval for mortality according to serum albumin concentrations.** ATT, anti-tuberculosis therapy; STM, streptomycin.

In a previous study from our cohort, we showed that adding STM to an intensified ATT could reduce mortality in HIV infected patients with TB meningitis (*Alvarez-Uria et al., 2015*). Given the poor penetration in cerebrospinal fluid of STM (*Kaojarern et al., 1991; Ellard, Humphries & Allen, 1993*), these findings were intriguing and suggested that the synergetic effect of STM and the iATT could be also beneficial for other forms to TB. The results of the present study confirmed this hypothesis. The use of the iATT regimen was not able to improve survival compared with the standard of care. However, when STM was added to the iATT, we observed a statistically and clinically significant mortality reduction.

In fact, the synergetic effect of STM and other anti-tuberculosis drugs is not new. In patients with pulmonary tuberculosis, the combination of rifampicin and STM has a strong bactericidal activity during the first six days of ATT (*Jindani, Doré & Mitchison, 2003*). In the same study, ethambutol, which was not used in our iATT regimen, had an antagonistic effect when combined with other drugs (*Jindani, Doré & Mitchison, 2003*). Moreover, in two phase II clinical trials, substitution of ethambutol with fluoroquinolones resulted in a more rapid decline in bacterial load and higher proportion of patients achieved TB culture negativity at eight weeks (*Rustomjee et al., 2008; Conde et al., 2009*). However, in our study, substitution of ethambutol with levofloxacin did not improve survival in the iATT group. New studies are needed to clarify if the synergetic effect on survival of STM and the iATT could be explained by the higher doses of rifampicin, the absence of ethambutol, the addition of levofloxacin or combinations of these factors.

On the other hand, a more rapid bactericidal activity during the first days of treatment might not be beneficial to all patients. In our study, patients with severe hypoalbuminemia did not benefit from the use of the iATT + STM regimen. These patients were likely to have more advanced forms of TB and a rapid killing of mycobacteria might have led to a

 

higher inflammatory response (*Ordonez et al., 2014*). Then again, hypoalbuminemia has been associated with lower concentrations of antimycobacterial drugs in patients with pulmonary TB (*Tappero et al., 2005*). New studies investigating new treatment approaches in this particularly vulnerable group are urgently needed.

The study has some limitations. Observational studies can be biased due to unknown confounders not included in the multivariate analysis. Unlike in clinical trials, treatment groups were not uniformly balanced. Patients in the iATT + STM group were more likely to be previously treated of tuberculosis, which is a known factor related to poorer prognosis (*Santha et al., 2002*; *Jones-López et al., 2011*; *Alvarez-Uria et al., 2012a*). However, in the sensitivity analysis using IPTW, being previously treated of tuberculosis was not included in the multivariate analysis and the use of iATT + STM remained statistically associated with improved survival. One of the most common side effect of STM is nephrotoxicity. It could be possible that the use of STM could have resulted in higher probability of acute kidney injury, but we did not collect information of renal function after ATT initiation, so we cannot compare nephrotoxicity among treatment groups. Nevertheless, we performed extensive analysis of baseline renal function with different parameters. Kidney dysfunction was associated with higher risk of death, but there was no interaction between baseline kidney dysfunction and STM use in terms of increased mortality. The study was done in a resource-limited setting, so we did not confirm TB diagnosis by mycobacterial culture. In addition, we did not have information about the drug resistance of mycobacteria among treatment groups. However, patients in the iATT + STM group were more likely to have been treated of TB in the past, which is a well-known factor for resistant TB. Then again, severely ill patients were not excluded, so the study reflects the "real-life" of HIV-associated TB in a resource-poor setting, so our results could be generalized to similar settings in developing countries with high prevalence of TB and HIV.

## CONCLUSIONS

The results of this study are consistent with previous studies from our cohort showing improved survival when STM is combined with an iATT regimen with higher doses of rifampicin and substitution of ethambutol with levofloxacin. However, the survival benefits of the new regimen was not evenly distributed among all patients. The new regimen did not reduce mortality in those patients with severe hypoalbuminemia. The study is observational in nature, so we should be cautious about our conclusions. However, given the high mortality of HIV-related TB, the results of this study deserve further research, ideally through a randomized clinical trial.

### Funding
The authors received no funding for this work.

### Competing Interests
The authors declare there are no competing interests.

## Author Contributions

- Gerardo Alvarez-Uria conceived and designed the experiments, analyzed the data, contributed reagents/materials/analysis tools, wrote the paper, prepared figures and/or tables, reviewed drafts of the paper.
- Manoranjan Midde and Praveen K. Naik contributed reagents/materials/analysis tools, reviewed drafts of the paper.

## Human Ethics

The following information was supplied relating to ethical approvals (i.e., approving body and any reference numbers):

The VFHCS was performed according to the principles of the Declaration of Helsinki, and was approved by the Ethics Committee of the Rural Development Trust Hospital (Reference number OS/003). Written informed consent was given by patients or caretakers for their information to be stored in the study database and used for research.

## Data Availability

Although the names of patients are not mentioned in the data, the datasets contain detailed information that could be used as indirect identifiers and could make it possible to discern the identities of HIV-infected patients included in the study. Therefore, in order to protect the confidentiality of the patients, we cannot place the data in a public repository.

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
