# Peer review of "Mortality in HIV-infected patients with tuberculosis treated with streptomycin and a two-week intensified regimen: data from an HIV cohort study using inverse probability of treatment weighting"

_PeerJ, doi:10.7717/peerj.2053_

## Round 0.1 · original submission · Major Revisions

All the reviewers' comments need to be addressed and the manuscript will be re-reviewed.

Reviewer 1 ·

Basic reporting

The hypothesis seems to be that streptomycin intensification of antituberculous therapy improves therapy for TB and leads to increased survival. There are no data for the TB response, only for mortality. The reader has to take the logical leap that the improvement in mortality that the authors report is due to improvement in TB. In fact there are no data presented regarding TB responses. The authors should report TB responses and causes of death for those who expired in the study

Experimental design

The authors compare mortality rates in three groups of individuals receiving different anti-tuberculous regimens. It is not sufficiently detailed how patients were assigned to the different groups.

Validity of the findings

no comments

Additional comments

In an observational single site cohort study, the authors investigated mortality in groups of individuals with tuberculosis treated with standard (ATT) or intensified antituberculous therapy, either with ciprofloxacin or with Streptomycin. The investigators found that there was a 28% reduction in mortality in individuals who underwent intensification with the addition of streptomycin. The data are interesting and the subject is of interest because therapy of tuberculosis in the setting of HIV remains a difficult therapeutic endeavor, especially in resource limited settings.
The authors describe three therapeutic groups; its seems clear that the patients were not randomized to these groups, but its not clear how the assignments were made. The ATT + STM group included individuals previously treated for tuberculosis, and one would have thought that these would be more difficult to treat.
It was surprising that albumin levels were different among the groups, with medians of 3.1, 3.15. and 3.2 – this seems quite unusual, can more be discussed to explain? Perhaps a box and whisker plot might be more persuasive? The authors refer to a severe hypoalbuminuria, but it is not clear to what this is referring, as the medians are not that much different. In addition, there is a strong emphasis in the text on individual laboratory values but it is not clear that there is a strong biological relevance to these statistically significant results (why is an elevated urea level associated with mortality?). In this regard, the ms could be improved by concentrating on the clinical outcomes.
The diagnosis of TB in these patients should be clarified- is this made by presumptive diagnosis, radiographic appearance, or sputum smear?

Reviewer 2 ·

Basic reporting

The manuscript “Mortality in HIV-infected patients with tuberculosis treated with streptomycin and a two-week intensified regimen: Data from an HIV cohort study using inverse probability of treatment weighting” by Alvarez-Uria et al, compares the mortality rate among HIV-infected individuals diagnosed with TB, and treated with three different regimens against the mycobacterial infection: (i) standard anti-tuberculosis therapy, (ii) intensified therapy and (iii) intensified therapy combined with streptomycin. The main conclusion of the manuscript is that adding streptomycin for two weeks to an intensified anti-TB regimen may reduce mortality.

Experimental design

The clinical data analyzed in this work was from an Indian HIV Cohort Study database, including 1615 patients that received anti-TB treatment

Validity of the findings

The findings are not new, and the main conclusion could be influenced by significant clinical factors other than adding streptomycin to the treatment.

Additional comments

The main conclusion is not really a new finding, as the authors acknowledge in the discussion. A more thorough analysis of the clinical data, including the impact of antiretroviral treatment in the survival of the patients diagnosed with TB is needed. In fact, low CD4 count is listed as a factor associated with increased mortality in this Cohort, which suggest that factors associated with the medical care of HIV infection could have a significant impact in the survival rate of the patients.
This manuscript will benefit including a broader discussion of the problems and approaches to treat TB infection in HIV patients from other areas of the world

Reviewer 3 ·

Basic reporting

It met criteria.

Experimental design

It met criteria.

Validity of the findings

It met criteria.

Additional comments

1. In the method section of abstract, the following sentence lines 28-30 are relatively ambiguous. Please separate the factors associated with higher mortality and those did with lower mortality.
2. Line 70: Please give more details on the definition of diagnosing TB in this study. Did all patients have positive acid-fast stains or positive mycobacterial cultures? Did the study exclude the patients who had culture yielding non-tuberculous mycobacterium?
3. Line 86: Did the study include TB lymphadenitis at all sites, such as intra-abdominal LN and intrathoracic LN.
4. Lines 89-95: Were the doses of all anti-TB drugs as well as streptomycin body weight-based or fixed? Please clarify them.
5. Line 95: Did the sentence “ART between 2 to 8 weeks after hospital discharge” mean all patients in the cohort received ART at that period?
 If yes, please give the reason why the factor “ On ART” were included in the univariate and multivariate analyses. The second question is that whether all patients needed hospitalization.
6. Table 2: was there high correlation among the factors “urea”, “BUN/Cr”, and eGFR in the multivariate analysis?
7. Please give more details on the smear conversion rates among the treatment groups.

---

## Round 0.2 · accepted · Accept

The revision has been reviewed and Reviewer 2 has agreed that the manuscript has addressed the issues raised.